# Dysregulation of mTOR Signaling after Brain Ischemia

**DOI:** 10.3390/ijms23052814

**Published:** 2022-03-04

**Authors:** Mario Villa-González, Gerardo Martín-López, María José Pérez-Álvarez

**Affiliations:** 1Departamento de Biología (Fisiología Animal), Facultad de Ciencias, Universidad Autónoma de Madrid, 28049 Madrid, Spain; mario.villa@uam.es (M.V.-G.); gerardo.martin@uam.es (G.M.-L.); 2Centro de Biología Molecular “Severo Ochoa” (CBMSO), Universidad Autónoma de Madrid/CSIC, 28049 Madrid, Spain

**Keywords:** mTOR, mTORC1, mTORC2, MCAo, brain ischemia, glia, microglia, astrocytes, oligodendrocytes, neuron

## Abstract

In this review, we provide recent data on the role of mTOR kinase in the brain under physiological conditions and after damage, with a particular focus on cerebral ischemia. We cover the upstream and downstream pathways that regulate the activation state of mTOR complexes. Furthermore, we summarize recent advances in our understanding of mTORC1 and mTORC2 status in ischemia–hypoxia at tissue and cellular levels and analyze the existing evidence related to two types of neural cells, namely glia and neurons. Finally, we discuss the potential use of mTORC1 and mTORC2 as therapeutic targets after stroke.

## 1. mTOR in the Brain under Physiological Conditions

### 1.1. The Structure of mTOR and Its Complexes in the Brain

Mammalian/mechanistic target of rapamycin (mTOR) is a 289 kDa serine–threonine kinase and a key element of two mTOR complexes called mTORC1 and mTORC2 (mTORCs) [1,2,3,4]. Furthermore, mTOR is highly conserved and is the center of multiples signaling pathways and coordinates important cellular processes such as cell growth and metabolism [5]. Although mTOR is ubiquitously expressed, it is especially abundant in the brain [6]. Therefore, mTOR dysfunction profoundly affects the central nervous system (CNS). Mutations in genes encoding mTOR regulators induce neurological disorders called “mTORopathies” [5].

Furthermore, mTOR, as indicated by its name, is a target protein of rapamycin, an immunosuppressant and anti-fungal macrolide compound isolated from *Streptomyces hygroscopicus*. This kinase comprises several functional domains, including C-terminal small FAT domain (FATC), C-terminal kinase domain (KD), FKBP12 rapamycin-binding domain (FRB), transactivation/transformation-associated domain (FAT), and an N-terminal domain containing at least 20 HEAT (Huntingtin elongation factor 3 A subunit of PP2A TOR1) repeats. The latter provide sites for the interaction of regulatory proteins to form mTORC1 and mTORC2. The KD domain of mTOR, with conserved sequences homologous to the catalytic domain of the phosphoinositide 3-kinase (PI3K) family, contains phosphorylation sites that regulate the activity of this kinase [7]. 

Rapamycin and its analogs (called rapalogs) act as allosteric inhibitors of mTORC1 by interacting with the FRB domain of mTOR via FKBP12 protein (FK506-binding protein 1 A 12 kDa) [6]. Furthermore, mTORC2 is insensitive to rapamycin inhibition, as initially described [8], but prolonged exposure to this macrolide results in the disruption of the assembly and integrity of mTORC2, thereby causing the functional inhibition of the complex [9].

Additionally, mTORC1 and mTORC2 share several common proteins, including the catalytic subunit mTOR, Deptor (DEP-domain-containing mTOR interacting protein), mLST8 (mammalian lethal with Sec13 protein 8), and Tti1/Tel2 complex [10] (Figure 1). In addition, each complex has specific proteins. Raptor (regulatory-associated protein of mTOR) and PRAS40 (proline-rich Akt substrate 40 kDa) are specific subunits of mTORC1, while Rictor (rapamycin-insensitive companion of mTOR), mSin1, and Protor1/2 are exclusive to mTORC2 [7] (Figure 1). All of these proteins have different functions in the complexes. Not only do they have structural functions (stabilizing the complexes and recruiting mTOR substrates) but they also contribute to regulating mTOR activity. Recently, an advancement in our understanding of the precise functions of each mTOR companion protein beyond the strictly structural function has occurred. Recent studies demonstrate an important impact in the fine-tuned activity of mTOR kinase, according to post-translational modifications of some of the companion proteins, mainly by phosphorylation. 

Deptor is an inhibitor of mTOR. The ablation of Deptor increases tumor size and causes the proliferation, migration, and invasion of tumoral cells because of the resulting overactivation of mTOR [11]. Furthermore, several post-translational modifications of Deptor can affect its inhibitory function. Recent studies show that the phosphorylation of Deptor at Tyr^289^ increases mTOR activity by preventing the correct coupling of the kinase to its protein-binding partner to form a complex [12,13]. This observation reveals the involvement of a novel molecular switch in the fine regulation of mTORCs.

The importance of mLST8 in mTORC activity is under debate. Studies using *Drosophila melanogaster* reveal that mLST8 is essential for mTORC2 activity, since LST8 knockout conserves mTORC1 but not mTORC2 activity [14]. It has been proposed that mLST8 specifically affects the interaction between mTOR and Rictor and is essential for the proper assembly of mTORC2 [15]. However, a recent study supports the notion that mLST8 is equally essential for ensuring the stability and activation of the two complexes through a mechanism that involves the kinase Akt [16]. PRAS40 is a negative regulator of mTORC1. Akt phosphorylation of PRAS40 increases the inhibitory effect on mTORC1, measured as an increase in autophagy [17]. Furthermore, mSin1 has been considered a scaffold protein with no relevance for mTOR activity [18]. However, recent data suggest that mSin1 is more than a structural protein. Indeed, it is essential for Akt phosphorylation at Ser^473^ by mTORC2 [19]. In addition, mSin1 defines the subcellular location of mTORC2, which also determines the final activity of the kinase. Alternative splicing of mSin1 generates five isoforms, of which at least three are able to assemble into mTORC2 and which determine three distinct mTORC2 complexes. These can be located in different subcellular compartments and their sensitivity to activation by PI3K differs [19]. 

From a functional perspective, mTORCsare considered molecular sensors of cellular energy status. They monitor several extra- and intra-cellular factors to orchestrate a response to maintain cellular homeostasis [20]. Furthermore, mTORC1 is involved in key cellular anabolic and catabolic functions, such as the synthesis of macromolecules (both lipids and proteins), cellular growth, autophagy, and cell cycle progression (Figure 2). In contrast, mTORC2 is related mainly to cell survival, as well as cytoskeletal organization and remodeling [21] (Figure 2). Therefore, both mTORCs are essential for cell viability, as evidenced using knockout mice for mTOR, Raptor, or Rictor. In all of these mouse models, embryo viability is severely compromised [22,23,24,25]. 

As both mTORC1 and mTORC2 regulate important cellular functions, their activity is finely regulated. Knowledge of the regulation of mTORC1 is currently much greater than that of mTORC2. Given that the latter is an important player in the maintenance of cell survival and may be an interesting target in neurodegenerative diseases, we consider that greater efforts should be devoted to unraveling the fine details of mTORC2 regulation. 

The principal mechanism of mTOR activation are related to post-translational modifications (mainly phosphorylation). The mTORCs have multiple regulatory phosphorylation sites, not only in the catalytic subunit (mTOR), but also in other subunits of the complex, such as Raptor, Rictor, Deptor, and PRAS40 [26,27,28]. The data available point to a relationship between the degree of phosphorylation of each subunit and the activity levels of mTOR. This notion opens up a new perspective of the fine regulation of mTOR activity, which could be modulated similarly to volume control and would differentially affect the phosphorylation levels of the substrates and their activities. It is important to keep this perspective in mind in the context of brain ischemia, since the energy status of cells changes rapidly and heterogeneously depending on the degree of injury [29]. In this ischemic scenario, fine adjustment of mTOR activity might be required to maintain a balance between anabolic and catabolic processes, thereby improving cell survival [30]. In addition, the subcellular localization of mTORCs is essential for their proper activity. In this regard, studies show that the activation of mTORC1 requires its localization in lysosome membranes (see Section 1.2.3.) [31]. 

### 1.2. Upstream Regulatory Pathways of mTORCs 

One of the most remarkable characteristics of mTORCs, especially mTORC1, is that these complexes are cellular energy sensors, and as such they act as signal convergence centers from extra- and intra-cellular “energetic factors”. The presence or absence of these factors modulates the final activity of mTORC1, allowing it to trigger distinct cellular responses to modify the balance between anabolism and catabolism, adjusting it to cellular needs (Figure 3).

Generally, the kinase activity of mTORCs promotes cellular anabolic pathways (translation, transcription, and lipid synthesis) and downregulates catabolism (protein degradation and autophagy). Interestingly, both complexes show a hierarchical position in the signaling pathways and are regulated by specific extra- and intra-cellular factors.

Furthermore, mTORC1 is regulated by multiple pathways related to the availability of trophic and growth factors, nutrients (especially amino acids and glucose), and oxygen (Figure 3). In contrast, mTORC2 activity is regulated mainly by growth factors, hormones, and neurotransmitters by the upregulation of PI3K activity [5,32]. However, knowledge about the regulation of mTORC2 in the CNS is scarce. Several studies show the importance of mTORC2 in cytoskeletal organization and remodeling and neuronal morphology. Neuronal mTORC2 is activated by neurotrophins, glutamate, NMDA, and inducers of long-term potentiation (LTP), the latter being a pivotal mechanism in synaptic plasticity and memory [32].

In this section, we describe the main regulatory pathways of mTORC1 and the principal factors that modulate the activation level of this kinase. 

#### 1.2.1. The Canonical Pathway: PI3K/Akt/mTORC1 and Growth Factors

This pathway senses the availability of several growth or survival factors, including nerve growth factor (NGF), brain-derived growth factor (BDNF), vascular endothelial growth factor (VEGF), insulin, insulin-like growth factor-1 (IGF-1), and neurotrophins (NT-1, -3, and -4). These molecules activate the PI3K/protein kinase B (PI3K/Akt) pathway by binding to their specific membrane receptors, which belong to the tyrosine kinase receptor superfamily (RTK), or to G-protein-coupled receptors (GPCRs) [20]. Full Akt activation requires its phosphorylation at two sites, namely Thr^308^ via the PI3K/Akt pathway and Ser^473^ via mTORC2 activity (Figure 3). First, Akt must be recruited to the cellular plasma membrane by direct interaction with phosphatidylinositol (3,4,5)-triphosphate (PIP3) [33]. The Ser^473^ phosphorylation site reveals the exquisite relationship between the activities of the two mTORCs, as mTORC1 activation requires full Akt activity, which in turn calls for previous mTORC2 activation [34]. Some studies show that Akt activity is finely regulated and that this protein kinase requires phosphorylation at other residues, such as Ser^477^ and Thr^479^, to enhance its interaction with mTORC2 and stability [33]. 

Akt activation induces the inhibition of TSC (Figure 3), a trimeric complex formed by TSC1 (hamartin), TSC2 (tuberin), and the scaffold protein TSC1D7. Several pathways converge on TSC to regulate (both positively and negatively) mTORC1 activity [27]. TSC is a GTPase-activating protein for the small GTPase Rheb (Ras homolog enriched in brain). Rheb is present in an inactive or activated state by binding to GDP or GTP, respectively [35]. Akt phosphorylates TSC2 at Thr^1462^, thereby disassembling and inactivating the complex. This process allows a Rheb-GTP-active state, which consequently induces mTORC1 through an unknown mechanism (Figure 3) [35,36]. The subcellular localization of mTORC1 in this step is essential for its full activation, as previously mentioned (see Section 1.1), since Rheb-GTP is located in the lysosomal membrane (see Section 1.2.3) (Figure 3). 

#### 1.2.2. AMPK–mTORC1 Pathway: The Glucose Sensor

AMPK, a negative regulator of mTORC1 activity, comprises three subunits (α, β, and γ), and it senses cellular energy status and glucose availability [37]. The highly energy-demanding brain is unable to store glucose and is greatly dependent on constant glucose supply from blood. A decrease in glucose supply induces a reduction in the AMP/ATP and ADP/ATP ratios, which is sensed by AMPK (Figure 3). This detection induces AMPK activation by phosphorylation at Thr^172^ of its α-subunit by LKB1 tumor suppression kinase [38]. Ca^2+^-activated Ca^2+^/calmodulin-dependent kinase β (CaMKKβ) and transforming growth factor-β-activating kinase 1 (TAK1) are both activators of AMPK [39,40]. 

The activation of AMPK inhibits mTORC1 activity through phosphorylation on two targets, namely TSC2 and Raptor [41]. Therefore, AMPK regulates mTORC1 activity at two levels (Figure 3). On the one hand, AMPK phosphorylation of TSC2 at Thr^1227^ and Ser^1345^ improves the stability of the TSC complex [35], promoting the inactive state of Rheb-GDP and inducing the inhibition of mTORC1 activity [42]. On the other hand, Raptor phosphorylation at Ser^722/792^ by AMPK disrupts mTORC1 and induces its inhibition [43]. AMPK activation and mTORC1 inhibition lead to autophagy (see Section 1.3). 

A reduction in oxygen levels, as occurs under hypoxia, decreases cellular ATP levels by inhibiting oxidative phosphorylation and other metabolic programs. This scenario promotes an ATP/AMP imbalance, inducing AMPK activation [44] and mTORC1 inhibition. 

#### 1.2.3. REDD1 and mTORC1

Reduced oxygen availability is another scenario that negatively modulates mTORC1 [45] (Figure 3). Hypoxia induces an increase in regulated DNA damage and development 1 (REDD1; also known as RTP801/DDIT4) expression, a highly conserved stress-response protein [46]. REDD1 downregulates mTORC1 through a TSC-dependent mechanism [47,48]. REDD1 triggers the release of TSC2 with the adapter protein 14-3-3, stabilizing the interaction between TSC1 and TSC2 and inducing mTORC1 inhibition [47]. Increased REDD1 expression early after ischemia has been described in neurons and glial cells [49].

#### 1.2.4. Regulation of mTORC1 by Amino Acid Levels 

Amino acids are key elements for neural cells. Moreover, amino acids are another regulatory pathway of mTORC1 involving a molecular mechanism that is not fully understood. Amino acids levels have an important influence on mTORC1 activity as they mediate the translocation of these kinases to the lysosomal membrane, a necessary step for it activation [21,36]. Rag GTPases have been reported to be involved in this process [21,36,50,51]. These molecules are small G-proteins that belong to the Ras superfamily andare present as heterodimers—RagA/B dimerized with RagC/D. The active conformation of these Rag heterodimers is RagA/B binding to GTP (RAG-A/B^GTP^) and RagC/D binding to GDP (RAG-C/D^GDP^) (Figure 3) [52]. Amino acid availability allows the active conformation of Rag, which binds directly to Raptor and induces the recruitment of mTORC1 to the lysosomal membrane [53] (Figure 3). At this location, mTORC1 is accessible to Rheb as a result of their proximity, and consequently mTORC1 is activated [30,53]. Leucine is a key amino acid that influences mTORC1 activity as it enhances the stabilization of the Raptor–mTOR interaction [54]. 

Cerebral ischemia is followed by a reduction in blood flow to the brain. This disruption causes the dysregulation of all upstream pathways that regulate mTORC1 (see Section 2). However, the impact of each individual upstream pathway on the activity of this complex is unknown. In this regard, it would be interesting to determine the relevance of each pathway for the final activity of mTORC1, which could differ between cell types. Preliminary data obtained in our laboratory using primary cultures of neurons and astrocytes suggest that the reductions in several energetic factors (glucose, oxygen, and trophic factors) have a summative effect on the activity levels of mTORC1.

### 1.3. Downstream Targets of mTORCs

The strategic position of mTORC1, downstream of three important signaling pathways, makes this complex an essential convergence center to check cell energy status. In this section, we describe the signaling pathways downstream of mTORC1, their principal targets, and the main cellular processes that they regulate. In addition, we provide the scarce details available for the downstream pathways of mTORC2 in the CNS.

One of the best known cellular processes regulated by mTORC1 is protein synthesis (Figure 4), which is essential for neural cell survival, synaptic plasticity, and brain development, and is dysregulated in several brain conditions, such as ischemia [27]. In neurons, trophic factors such as BDNF, insulin, and IGF-1, as well as some neurotransmitters, induce an increase in protein synthesis by local mTORC1 activation [55,56]. Protein synthesis is regulated by two well-characterized targets of mTORC1, called p70 ribosomal protein S6 kinase (P70S6K) and eukaryotic initiation factor 4E (eIF4E)-binding proteins (4EBPs), which trigger elongation and initiation of protein translation, respectively [57] (Figure 4). Furthermore, mTORC1 phosphorylates 4EBPs at several residues, thereby allowing the release of a group of eukaryotic initiation factors (eIFs) located on 5’-UTR of cap-dependent mRNA that initiates translation [58,59]. Three isoforms of 4EBP have been described, namely 4EBP-1, 4EBP-2, and 4EBP-3, with 4EBP-2 being the most common in the CNS [59]. Additionally, mTORC1 phosphorylates 4EBPs mainly at Thr^37/46^ and Ser^65^. It has been proposed that hierarchical phosphorylation of 4EBPs may be crucial in the fine regulation of translational processes mediated by mTORC1 [60,61,62]. P70S6K is the other main substrate of mTORC1 related to protein synthesis. Additionally, mTORC1 phosphorylates P70S6K at Thr^389^, thereby allowing the recruitment of the 40S ribosomal subunit to the translational machinery. Furthermore, P70S6K phosphorylates eukaryotic initiation Factor 2 kinase (eIF2K), which induces the elongation phase of protein synthesis [21].

The synthesis of structural lipids allows the length of the plasmatic membrane to increase, a key aspect in axonal growth, dendritic arborization, and myelination (Figure 4). In addition, lipids are important molecules in cellular metabolic processes, since they are sources of energy, especially when glucose is lacking, as occurs after ischemia [63,64,65]. Furthermore, mTORC1 inhibition using rapamycin downregulates lipid synthesis. This observation, reveals that mTORC1 participates in this process [66]. Additionally, mTORC1 activates sterol regulatory element-binding proteins 1 and 2 (SREBP1-2) through P70S6K, since ablation of P70S6K induces reductions in the amount of lipid synthesis and cell size [67]. Activated SREBP is translocated to the nucleus to act as a transcription factor of genes involved in lipogenesis [68,69]. In the developing brain, mTORC1 induces the transcription of numerous genes related to the sterol–cholesterol biosynthesis pathway. Altered expression of these genes by dysregulation of mTORC1 contributes to neurodevelopmental disorders [70] and also affects myelination [71], an essential process for neuronal recovery after cerebral ischemia (see Section 2.2.3). 

Autophagy is a highly conserved cellular catabolic mechanism that involves the degradation of damaged cellular components, misfolded proteins, long-lived proteins, and damaged organelles by lysosomes. It is believed that after injury, autophagy plays a critical role in removing damaged molecules and subcellular components to maintain cellular homeostasis [72]. Furthermore, mTORC1 is a key player in the regulation of autophagy (Figure 4). It induces this catabolic mechanism and facilitates the fusion of the autophagosome with the lysosome, a key step in this process [73]. In nutrient-rich conditions, active mTORC1 inhibits autophagy by phosphorylating Unc-51-like kinase 1 (ULK1) or UV radiation resistance-associated gene protein (UVRAG), among others [74,75,76,77,78]. Inversely, in nutrient-poor conditions, reduced mTORC1 activity induces autophagy, which leads to the removal of proteins and organelles to compensate for nutrient starvation. As we described previously, a reduction in glucose levels induces AMPK activation, which phosphorylates Raptor, which in turn inhibits mTORC1 and triggers autophagy [73,79,80]. Inactivation of mTORC1 under starvation conditions prevents the maintenance of Ser^757^ phosphorylation of ULK1, which induces autophagy initiation [81]. As a general idea, autophagy induction after ischemia could be considered an intrinsic mechanism of neuroprotection. In this regard, some experimental evidence confirms this hypothesis [79]. The pharmacological induction of autophagy after ischemia using rapamycin reduces brain damage [73,82,83,84]. However, other results indicate that excessive autophagic flow aggravates ischemic damage [85]. Therefore, autophagy is another example of a cellular mechanism that requires fine regulation to have positive or negative effects after injury. 

The mTOR activity plays a pivotal role in axonal growth, a highly regulated process involved in synaptic plasticity and development [86]. Rapamycin administration to primary cultures of neurons induces the inhibition of axonal growth and prevents neuronal differentiation [32]. However, mTORC1 activation by ablation of their negative upstream regulators (TSC or PTEN) stimulates regenerative processes related with axon guidance and growth [87,88]. In this regard, the presence of some components of the mTORC1 pathway, including P70S6K and 4EBP-1, has been reported in the axonal cones of primary neurons. This interesting observation reflects the importance of mTORC1 location in the appropriate subcellular compartment to regulate certain neuron-specific mechanisms, such as axonal growth, in situ [26]. 

The downstream targets of mTORC2 include several members of the AGC kinase family, such as Akt, PKC, and SGK1 [89] (Figure 4). As mentioned previously, mTORC2 participates in neuronal cytoskeleton remodeling, specifically in the actin cytoskeleton [8]. Rictor ablation induces a reduction in mTORC2 activity, thereby having impacts on neuronal size and morphology [90,91]. Indeed, cytoskeleton rearrangement is a crucial factor for synaptic plasticity [92]. 

## 2. mTOR after Cerebral Ischemia

Ischemic stroke is a devastating disease induced by partial or total occlusion of a cerebral artery, the middle cerebral artery being that most frequently affected in humans [93]. Stroke is now the leading cause of disability and the second cause of death worldwide [93,94]. Ischemia accounts for 80% of all strokes.

The occlusion of a cerebral artery dramatically reduces the blood flow to several brain regions (Figure 5). As a result, there is a decrease in the supply of nutrients (glucose), oxygen, and growth factors to neurons. These conditions lead to neuronal injury or death, bringing about serious neurological dysfunction, the severity of which depends on the size of the area affected in the brain and the duration of the occlusion [95].

To date, the only therapeutical intervention available to ameliorate ischemic damage is a reduction in neuronal injury via reperfusion using a thrombolytic agent (recombinant tissue plasminogen activator, tPA) or surgical removal of clots [84]. However, these strategies are very limited because they are only feasible within a short time window after ischemia onset due to the high risk of intracerebral hemorrhage, which would worsen the prognosis of the patient. Given these considerations, reperfusion is suitable for only a few patients (approximately 5%) [96]. Although reperfusion can effectively decrease the infarct volume and improve neurological recovery, it does not reduce the subsequent neurodegeneration that occurs after brain ischemia. Moreover, ischemia–reperfusion injury occurs, consisting of multiple pathological events, such as excitotoxicity, oxidative stress, inflammation, apoptosis, and blood–brain barrier (BBB) disruption [97]. In this regard, it is necessary to develop new therapeutic strategies to alleviate ischemic damage in situations with or without reperfusion.

Most of the neuroprotective compounds that have been effective in animal models are ineffective in humans. This has led to doubts about the suitability of in vivo models of stroke. However, we consider that the differences in the responses of animal models and humans to these agents reflect the complex mechanisms that underlie cerebral ischemia and highlight the need for a greater knowledge of the same. Along these lines, a deeper understanding of the molecular mechanisms triggered after cerebral ischemia will provide better opportunities to discover new therapeutic targets. In this regard, mTOR emerges as a potential candidate.

Two distinct approaches are used to study the effects of ischemia on the brain. The most common is an in vitro model, which involves primary cultures of neural cells exposed to oxygen and glucose deprivation (OGD). According to the literature available, the duration of OGD fluctuates from 30 min to 6–12 h. However, the percentage of oxygen remains more constant between the different studies, with 1% being the most common value used. Differences in the exposure to OGD could explain the observed discrepancies with respect to the role of mTOR in this model of ischemia. This in vitro model allows the study of the individual cellular and molecular changes that take place in each type of brain cell after ischemia. Nevertheless, the model cannot be used to analyze the relationship between cell types. 

The second type of approach used to examine the effects of ischemia on the brain involves in vivo model. They can be divided into two main groups on the basis of the extent of the affected tissue. The first group is global ischemia (two- or four-vessel occlusion), which reproduces the impact of a cardiac arrest on the brain and affects the entire brain. The second group, called focal cerebral ischemia, mimics occlusion of a cerebral artery either by an embolus or local thrombosis, with middle cerebral artery occlusion (MCAo) being the most frequent [84,98,99]. MCAO models can be subdivided into two groups: transient MCAo (tMCAo) and permanent MCAo (pMCAo). The tMCAo is when the occlusion of the artery is sustained for a short period (between 30–120 min) and reperfusion is allowed after. In pMCAo models, the occlusion of MCA is maintained until animal sacrifice. The tMCAo mimics a therapeutic intervention (reperfusion) in stroke patients, while pMCAo does not reproduce a therapeutic intervention. 

Ischemic brain injury is the result of a complex sequence of pathophysiological events that progress over time called the “ischemic cascade”. The main pathogenic mechanisms that evolve include excitotoxicity, peri-infarct depolarization, inflammation, and programmed cell death (apoptosis) (Figure 5) [99]. The affected region is not damaged homogeneously after focal ischemia. The area severely affected by hypoperfusion, named the ischemic core, presents a high rate of neuronal death by necrosis (Figure 5). The less affected region surrounding the core is called the penumbra and is characterized by apoptosis, which can be rescued (Figure 5) [95]. Given these considerations, one of the most important targets in neuro-regenerative approaches after ischemia is the penumbra region. The duration of occlusion determines the severity of damage and the prognosis of the patients [100]. The number of surviving neurons after a stroke defines cerebral functional deficits. In this regard, protecting neurons from death has been the focus for recovering cerebral functions after ischemia. Several approaches to achieve this goal include strategies to prevent neuronal death (neuroprotection) and others directed at neuronal repair; that is, neuron-based approaches. The current understanding of glial cell responses after cerebral ischemia reveals that these cells offer an interesting alternative strategy to protect or recover neurons and brain function.

The blockage of or reduction in blood flow to a brain region causes a drastic depletion of energy supply (glucose, oxygen, and growth factors), thereby leading to “energy failure” (Figure 5). All upstream modulatory signaling pathways of mTORCs activity (see Section 1.2) are partially or totally affected by this failure, and consequently the dysregulation of mTORC1 and mTORC2 ensues. 

At the tissue level, mTORC2 activity, measured by phosphorylation levels of its main substrate, pAkt-Ser^473^ (Figure 4), decreases in response to ischemia. Sustained mTORC2 inhibition in pMCAo models has been described, even days after the insult [101,102]. A decrease in mTORC1 activity, measured as a reduction in phosphorylation levels of its major targets P70S6K and 4EBPs (see Section 1.3 and Figure 4), has also been observed in pMCAo models [103,104]. Several studies have revealed a positive neuroprotective effect of mTORCs upregulation after cerebral ischemia [105,106]. In the tMCAo model, mTORC1 and mTORC2 activity suppression using Rapalink-1, a third-generation mTOR inhibitor, has been reported to exacerbate neuronal damage induced by ischemia in the short-term, worsen BBB stability, and increase the area of damage [107]. These observations suggest that the upregulation of mTORC activity after cerebral ischemia has beneficial effects on the CNS. 

Many studies on mTOR and cerebral ischemia have focused on autophagy, which plays a critical role in the maintenance of proteostasis and the survival of neurons by promoting the lysosome-driven removal of damaged or non-essential molecules and defective organelles (see Section 1.3). Canonically, autophagy is activated by starvation conditions, and consequently it induces the elimination of organelles and molecules to compensate. Autophagy is a key mechanism to protect neurons against ischemia. Treatment with rapamycin after MCAo decreases mTORC1 activity and increases autophagy, thereby reducing neuronal apoptosis [83,104]. In animal models, preconditioning with rapamycin improves brain tolerance to ischemic damage [108], ameliorates neurological deficits, and reduces infarct volume and brain edema [109,110,111]. Rapamycin administration before or after tMCAo reduces infarct volume and neurological dysfunction [83]. However, the role of autophagy in cerebral ischemia remains controversial, since current lines of evidence suggest that overactivation of autophagy induces cell death and aggravates ischemic brain injury [73,85]. Several data from in vivo models support the notion that autophagy suppression through the AMPK/mTORC1 pathway after ischemia–reperfusion (tMCAo models) reduces cerebral damage and brain edema, improving the neurological score [112,113,114]. This overactivation of autophagy is probably triggered by a dramatic decrease in mTORC1 activity through the upregulation of AMPK and downregulation of the PI3K/Akt pathway induced by the ischemic conditions. Post-ischemic treatment using specific autophagy inhibitors significantly decreases the volume of the injured area in global ischemia [115]. Furthermore, the reduction in autophagy through mTORC1 activation has proven beneficial in tMCAo models, in which reperfusion is allowed and the availability of nutrients and oxygen is restored. It would be interesting to analyze the role of autophagy in pMCAo models, in which the reestablishment of normal conditions does not occur. 

The effects of cerebral ischemia are far from homogeneous. It should be noted that cerebral ischemia triggers the activation of the mTORC1 pathway in the ischemic penumbra and a decrease in the ischemic core [83]. This is an important consideration to understand the effects of rapamycin treatment on in vivo models of ischemia, as the ischemic core and penumbra regions change over time.

Therefore, available data indicate that both an increase in mTORC1 activity and its inhibition by rapamycin induce beneficial effects after ischemia. These contradictory results lead us to advocate for further research into the role of mTORC1 in the ischemic process, perhaps taking into account the different models used and the duration of ischemia. Additionally, it is imperative to study the participation of mTORCs in each neural cell type. Rapamycin is a conventional mTOR inhibitor that especially affects mTORC1. However, after prolonged treatment, it also affects mTORC2 [9]. Experiments with rapamycin show both beneficial and detrimental effects on ischemia models. However, a meta-analysis indicated an overall neuroprotective effect of this macrolide when administer at low doses. Given this consideration, rapamycin may be effective as a therapeutic agent to treat ischemia [116].

### 2.1. mTORCs and Neurons

Neurons are the neural cells that are most sensitive to ischemic damage. Most studies addressing cerebral ischemia have focused on attempts to reduce neuronal death to ameliorate damage. Unfortunately, the promising results obtained in animal models using neuroprotective agents have not been reproduced in humans [95]. 

The role of mTORC1 and mTORC2 in neurons under ischemic conditions has been widely studied [20,117,118]. As we described previously, mTORC1 activity is finely tuned by multiple upstream signaling pathways (see Section 1.2). Ischemia dramatically reduces mTORC1 activity in neurons through the dysregulation of all of its upstream pathways (Figure 3) [34,103,119,120]. Many studies suggest that the canonical PI3K/Akt pathway is the most influential with respect to reducing neuronal mTORC1 activity after ischemia [121,122]. Ischemia promotes the depletion of growth and extracellular survival factors that downregulate the PI3K/Akt pathway, thereby inhibiting mTORC1 and triggering a reduction in phosphorylation levels of neuronal P70S6K, affecting protein synthesis and neuronal viability (Figure 6) [123,124,125]. Thus, recovery of mTORC1 activity through the activation of the neuronal survival PI3K/Akt pathway reduces the loss of neurons [101,102,103,126,127,128,129,130]. Additionally, the exposure of primary neurons to AZD2014, an mTOR inhibitor, after OGD induces an increase in neuronal death via the downregulation of the TSC/mTORC1 pathway [131]. 

Rapamycin is the most commonly used inhibitor of mTOR and a known autophagy inductor. An increase in autophagy or autophagic flux by rapamycin reduces neuronal apoptosis after OGD or reperfusion injury [119,132]. Zhang et al. [132] proposed a complex relationship between neuronal apoptosis and autophagy that could explain the beneficial effects observed with rapamycin. 

These seemingly contradictory results regarding the positive effects of either activation or inhibition of mTOR in neurons in the context of ischemia may be explained by the doses and time points selected for the analysis. Moreover, the signaling pathways chosen to impact mTOR activity could represent a source of high variability in the outcome. Related to this, upregulation of the survival pathway PI3K/Akt is known to have beneficial effects not only through mTORC1 activation, but also by modulating other relevant targets described to improve neurons survival, such as CREB, FOXO, or GSK3, among others [133,134]. On the other hand, a direct inhibition of mTOR by rapamycin both pre- and post-OGD improves neuronal viability via induction of autophagy. However, only low doses of rapamycin have been shown to be beneficial, while high doses prove toxic, probably by affecting mTORC2 activity and reducing neuronal survival [135].

Little information is available regarding mTORC2 activity in neurons. This complex is downregulated after brain ischemia, measured as a reduction in the phosphorylation levels of Akt-Ser^473^ [20,103,136], while increases in its activity have beneficial effects after damage [137]. 

### 2.2. mTORCs and Glial Cells

Glial cells are abundant in the mammalian CNS. However, their role in physiological and pathological conditions is not yet fully understood. As we gain a greater understanding of glial cells, their relevance increases. In this context, these cells emerge as potential therapeutic targets. However, limited data are available to support a relationship between glial cells and the activity of mTORC1 and mTORC2.

#### 2.2.1. Astrocytes 

Astrocytes are the most abundant cell type in the CNS, accounting for 50% of the human brain volume [138]. There are two main groups that differ in morphological and spatial distribution: fibrous astrocytes, which are predominant in white matter; and protoplasmic astrocytes, predominant in gray matter [139]. The functional role of each type after ischemic injury is unknown, so the term “astrocyte” is used herein to include both types. Astrocytes play an essential role in multiple physiological and pathological processes of the CNS, beyond the initial notion that they are mere supportive cells to maintain functional neurons and neuronal circuits. 

After cerebral ischemia, astrocytes have been associated with several positive effects, such as BBB stability, neuronal metabolic support, and neuroprotection. One well-known mechanism of astrocyte-driven neuroprotection is the capacity of these cells to reduce excitotoxicity by decreasing excessive extracellular glutamate released by damaged neurons. This decrease is achieved through the activation and overexpression of various astrocytic glutamate transporters, such as glutamate transporter-1 (GLT-1) [95]. In addition, astrocytes are involved in the management of oxidative stress [140]. 

The release of distinct inflammatory factors by ischemic neurons and reactive microglia induces astrocytes to switch to a reactive status, which can have a negative or positive effect on ischemic damage depending on the phenotype they acquire. Like microglia (see Section 2.2.2), astrocytes can acquire an A1 (pro-inflammatory) or A2 (anti-inflammatory or neuroprotector) phenotype. Reactive astrocytes reduce the detrimental accumulation of reactive oxygen species (ROS) during ischemia, thereby reflecting their potential role as anti-oxidative players under pathological conditions [95,141,142]. While there is extensive information about the role of astrocytes after cerebral ischemia, few studies have described the participation of mTOR activity in this context. Data currently available on the positive or negative role of mTORCs in astrocytes after cerebral ischemia are contradictory. 

Some studies show the beneficial effects of mTOR activation on astrocytes after OGD (Figure 6). This upregulation of mTORC1/S6K1 contributes to astrocyte survival after ischemia [125]. In addition, an increase in astrocytic mTOR activity after OGD through the activation of the PI3K/Akt pathway triggers the astrocytic release of VEGF and BDNF, which promote angiogenesis and enhance neuron survival [143]. Furthermore, mTOR activation has been reported to increase GLT-1, which promotes glutamate uptake and reduces the excitotoxicity of neurons [144,145,146]. All of the above-mentioned data suggest that mTOR activation in astrocytes has a positive effect within the context of ischemic damage. 

However, a beneficial effect of astrocytic mTOR inhibition under ischemic conditions has been demonstrated (Figure 6). The reduction in astrocytic mTORC1 activity via AMPK [147,148,149] or TSC2 [150] after OGD induces an increase in autophagic flow and a decrease in the release of pro-inflammatory cytokines [151,152], thereby improving neuronal viability. 

There are limited data regarding the role of astrocytic mTORC2 after ischemia. The increase in ROS after acute prenatal hypoxia triggers the inhibition of mTORC2 activity by ubiquitination and degradation of Rictor. This inhibition then leads to the inefficient differentiation of astrocytes, thereby affecting their response to ischemia and worsening damage to the brain [153]. After OGD, full mTORC2 activity is required for the overexpression of GLT-1, which is beneficial after ischemia by reducing excitotoxicity [145]. 

#### 2.2.2. Microglia 

Microglia are the first line of defense against cerebral damage and infection by pathogens. These cells are uniformly distributed throughout the brain [4] and they account for 10–15% of all cells in this organ. Under physiological conditions, microglia are in a “resting state”, using their highly motile processes to check the microenvironment of nervous tissue to generate an appropriate response and maintain tissue homeostasis [154]. After cerebral ischemia, this homeostasis is disrupted and microglia respond. 

After 2 days of pMCAo, the inflammatory stage is established. Ischemia-induced cell death results in the release of damage-associated molecular patterns (DAMPs), thereby activating microglia to acquire a reactive phenotype [84,101,136,155]. Given the phagocytic capacity of microglia as scavengers, they are the first glial cells to respond to cerebral ischemia [156]. The activation of microglia involves three processes, namely morphological transformation, migration to the damaged area, and proliferation [136]. 

Two phenotypes of active microglia, namely M1 and M2, have been identified. The M1 phenotype presents a pro-inflammatory profile and can release inflammatory cytokines such as IL-1 and TNFα. As a general concept, this phenotype exacerbates the inflammatory response and ischemic damage [95]. Accordingly, the inhibition of microglial activation by minocycline has been shown to protect the brain against focal ischemia by reducing the expression of IL-1β-converting enzyme and cyclooxygenase-2 [157]. M2 microglia release anti-inflammatory cytokines such as IL-10 and TGF-β, thereby promoting BBB stability and allowing functional recovery of the brain after ischemia [158,159]. An imbalance towards the M1 phenotype is related to poorer clinical prognosis [160]. In this context, most therapies based on microglia are focused on polarizing them towards the M2 phenotype to minimize the deleterious effects of the M1 phenotype [95,161,162]. Despite this classification, many studies support the notion of great heterogeneity in the population of activated microglia and the coexistence of intermediate phenotypes. This complexity hinders advances in our understanding of microglia after damage [163]. 

The mTORC1 pathway has recently been reported to be involved in microglial activation and their polarization towards the M1/M2 phenotype after ischemia. Furthermore, mTORC1 is a known regulator of immune responses. Some studies addressing mTORC1 inhibition after ischemia reveal promising results, including the reduction in the ischemic area and the amelioration of neurological deficits in animals. The direct inhibition of mTORC1, using pharmacological (sirolimus and everolimus treatment for 6 h after tMCAo induction) and genetic (Raptor-KO mice) approaches, leads to a reduction in pro-inflammatory cytokine expression, in parallel to an increase in the M2 phenotype [164]. Indeed, recent studies show similar results using different approaches to reduce mTORC1 activity, including the downregulation of some upstream players of the PI3K/mTORC1 pathway [165,166,167,168]. The mTOR activity regulates the synthesis of pro-inflammatory cytokines (Figure 6) through the mTOR/STAT3 pathway but also enhances autophagy in microglia [166,169]. Using tMCAo rats, it has been shown that early downregulation of the Akt/mTOR/STAT3 pathway after ischemia suppresses microglia-mediated neuroinflammation and improves cerebral injury [165].

#### 2.2.3. Oligodendrocytes 

Oligodendrocytes (OLGs) are responsible for axonal myelination in the CNS. Myelination, which occurs during embryonic development and continues into adulthood, determines the speed of nerve impulse conduction and supports axons [170]. Myelination is a complex and organized process with high metabolic demands [171]. In this regard, mTORC1, a pivotal player in the coordination of cell metabolism, is involved in this process. Some studies using both in vitro and in vivo models have demonstrated that mTORCs are fundamental players in myelination and OLG differentiation [3,172,173]. Several in vivo approaches using loss-of-function transgenic mice have unveiled the role of mTORC1 and mTORC2 in myelination. Knockout mice for Raptor or Rheb1, in which mTORC1 activity is abolished, show hypomyelination [3,71,172,174]. The same phenotype was observed in Rictor knockout mice, which show suppressed mTORC2 activity [3]. Ablation of mTOR, which affects the function of both complexes, also induces hypomyelination in the CNS [175]. This observed hypomyelination effect does not occur equally in all brain regions, with the cerebellum and spinal cord being more sensitive to mTORC ablation than other regions [176]. We conclude that the combined activity of mTORC1 and mTORC2 is necessary to maintain adequate myelination levels. Of note, the loss of mTORC1 function has a greater impact on myelination than the ablation of mTORC2, and simultaneous inactivation of the two complexes has a summative effect compared to the disruption of mTORC1 alone [172]. 

Like other cell types, in OLGs, mTORC1 regulates protein and lipid synthesis, both necessary processes for correct myelin synthesis. The inhibition of mTORC1 induces a reduction in the synthesis of myelin proteins by OLGs, thereby impairing myelination [3]. Conversely, depending on the upstream pathway affected, mTORC1 activation in OLGs induces a hyper- or hypo-myelinated phenotype. The overstimulation of Akt in OLGs triggers mTORC1 activation and enhances myelination, with no changes in the proliferation or survival of oligodendrocyte precursor cells (OPCs) or mature OLGs [177,178]. However, the overactivation of mTORC1 through deletion of the inhibitory complex TSC causes hypomyelination, which can be reversed by the administration of rapamycin [172,179]. In line with these results, the brains of patients with tuberous sclerosis (a disorder caused by loss of function mutations of TSC) show impaired white matter integrity [180]. The mechanism underlying this hypomyelination is not understood. It has been reported that mTORC1 overactivation by TSC ablation triggers a reduction in mTORC2 activity, as indicated by lower mTORC2-dependent phosphorylation levels of p-Akt-Ser^473^ [176,181]. Furthermore, mTORC2 has been associated with lipid metabolism [63] and specifically with the biosynthesis of sphingolipids, an abundant constituent of myelin [182]. This observation could explain the stronger defects in myelination observed in the aforementioned OLG double-Raptor^−/−^ and -Rictor^−/−^ mutants [172].

In addition to their participation in myelination, mTORCs have significant involvement in the differentiation and maturation of OLGs. Using in vitro models, it has been demonstrated that an increase in mTORC1 activity induces the differentiation of OLGs from OPCs. Conversely, the inhibition of mTORC1 using rapamycin prevents OLG differentiation [173,183]. In vivo models reinforce these findings. Conditional ablation of Rictor or Raptor has a differential impact on OLG differentiation. Furthermore, mTORC1 is a positive regulator of OLG maturation through an unknown mechanism, since ablation of Raptor induces an increase in OPC numbers, a selective decrease in myelin protein, and a reduction in the number of mature OLGs in the corpus callosum [3,176]. In contrast, Rictor ablation has a modest effect on OLG differentiation [172]. 

Cerebral ischemia induces a dual response in the OLG population. During the acute phase, a loss of OLGs occurs in the damaged area, since these cells are highly sensitive to oxidative stress, which occurs in this early stage of injury [184]. Some authors correlate the loss of OLGs and demyelination with a reduction in PI3K/Akt/mTORC1 activity (Figure 6), since the activation of this pathway via downregulation of PTEN enhances myelination and improves neuro-functional recovery from stroke [185]. In the long term, ischemia induces an increase in the number of OLGs, mainly in the penumbra [186,187]. This effect has been related to the activity of the PI3K/Akt/mTOR pathway [188]. Thus, an increase in the activity of this pathway after ischemia reduces the loss of OLGs and myelin, thereby improving neurological deficits (Figure 6) [185,189,190]. 

The “oligovascular unit”, which defines a dynamic structural complex composed of OPCs and endothelial cells (ECs), is gaining relevance. Interactions between OPCs and ECs play a pivotal role in angiogenesis in both physiological and pathological conditions [191]. After ischemia, the close communication between ECs and OPCs could be directed to recover myelin in the damaged area. In vivo treatment using EC secretomes induces beneficial effects on white matter, such as enhanced vascularization, induction of myelinization, and increases in the number of mature OLGs, thereby improving cognitive function [192]. The ischemia-induced proliferation of OPCs in the oligovascular unit is mediated by the PI3K/Akt/mTOR pathway [193]. 

## 3. Conclusions 

It is known that mTOR plays a pivotal role in coordinating antagonist cellular processes such as viability–apoptosis and anabolism–catabolism. Given the relevance of this kinase, its activation levels must be finely regulated. Furthermore, mTOR acts as a catalytic subunit of two complexes, named mTORC1 and mTORC2, regulated by multiple signaling pathways by the presence of energetic factors. Various lines of evidence highlight that post-translational modifications (mainly phosphorylation) of protein-binding partners of mTOR have a crucial impact on the final activity of these two complexes. 

In an ischemic situation, the loss of oxygen, glucose, amino acids, and survival factors triggers the dysregulation of all upstream pathways that modulate mTORCs, promoting marked reductions in mTORC2 and mTORC1 activities (Figure 7). Thus, pharmacological activation of these complexes emerges as a potent neuroprotective tool against ischemic damage, a pathological condition that has no effective treatment. Most in vivo results relate to the acute phase of the disease, limited to 60–90 min of ischemia and then reperfusion. This approach allows analysis of the effect of reperfusion after ischemia but not the effect on a long-term ischemic condition. The latter scenario is the most common situation in humans, either because the reperfusion is not allowed or it is performed 6 h after the onset of the first symptoms. This fact could explain the failure to translate the promising therapies in animal models to humans. In contrast to animal models, the time elapsed from the start of the artery blockage to the first symptoms of the disease in humans is not known. This could be an important factor to take into account when choosing targets for human therapies. 

Although animal models reveal the positive effects of rescuing the activity of mTORCs on damaged tissue, the inhibition of mTOR activity by rapamycin is also beneficial (Figure 7). This apparent contradiction indicates a need to devote further research efforts to unraveling the role of each mTORC in neural cells or the temporal evolution of mTORCs after ischemia. In vitro analysis, the approach to study the effects of ischemia on each distinct neural type, indicates various contradictions, especially regarding mTORC1. Given that available data on mTORC2 in the CNS and after ischemia are scarce, greater research efforts should be devoted to this promising field.

In astrocytes, there is consensus about the positive effect of mTORC2 activation after ischemia. In contrast, there are contradictory data about the precise role of mTORC1 in this cell population (Figure 7). These data may reflect the rapid astrocytic response after ischemia and point to the relevance of considering the duration of damage (the most variable parameter in OGD models). In addition, it is well known that the astrocytic population shows heterogeneity in both physiological and pathological conditions. As previously mentioned, after hypoxic–ischemic conditions, astrocytes switch to an activated state, which includes at least two distinct phenotypes, if not more. These have not been considered in studies related to mTOR to date. Available data regarding microglia are homogeneous and suggest that a decrease in mTORC1 activity after ischemia reduces neuroinflammation and increases M2 microglial phenotype, thereby ameliorating damage (Figure 7). The information about OLGs and mTORC1 reveals that this complex is a key player in the myelination and differentiation of these cells. After cerebral ischemia, demyelination occurs, and therapies to restore the myelin of damaged axons emerge as promising approaches to ameliorate ischemic damage in the long term (Figure 7). In this regard, the upregulation of the mTORC1 pathway after ischemia reduces the loss of OLGs, improves tissue myelination, and mitigate damage (Figure 7). 

Thus, after cerebral ischemia, the upregulation or downregulation of mTORC1 activity has beneficial effects, which probably depend on the temporal evolution of injury (Figure 7). In the short term (the first several hours), it is important to activate mTORC1 to improve neuron and OLG survival. However, when neuroinflammation is well established, the inhibition of mTORC1 reduces brain damage (Figure 7). These observations highlight the importance of understanding not only the pathological events that occur after ischemia, but also the temporal dynamics of mTORC1 and mTORC2 in response to ischemia.

## Figures and Tables

**Figure 1 ijms-23-02814-f001:**
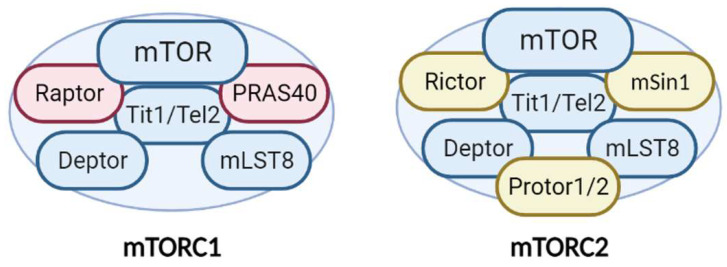
**Structure and components of the mTORC1 and mTORC2 complexes.** The mTORCs share common proteins (blue) called mTOR (catalytic subunit), Deptor (mTOR inhibitor subunit), mLST8 (scaffold and activator subunit), and Tit/Tel2 (assembly subunit). The specific mTORC1 proteins (pink) include Raptor and PRAS40 (mTOR inhibitor). Rictor, mSIN1, and Protor1/2 (activity modulator) comprise mTORC2 (yellow). See text for more details.

**Figure 2 ijms-23-02814-f002:**
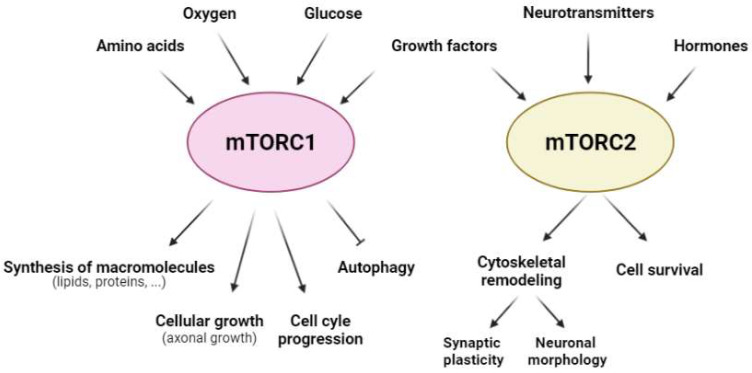
**Regulatory factors of mTORC1 and mTORC2 and the cellular responses that they govern.** The activity of mTORC1 (pink) and mTORC2 (yellow) is modulated by a range of extra- and intra-cellular factors; mTORC1 (pink) is regulated mainly by amino acids, oxygen, glucose, and growth factors, whereas mTORC2 (yellow) is dependent on growth factors, neurotransmitters, and hormones in the CNS. Furthermore, mTORC1 regulates anabolic and catabolic cellular processes, such as the synthesis of macromolecules (lipids and proteins), cellular growth, cell cycle progression, and autophagy; mTORC2 modulates cellular processes that involve cytoskeletal remodeling, such as synaptic plasticity and neuronal morphology, as well as cell survival.

**Figure 3 ijms-23-02814-f003:**
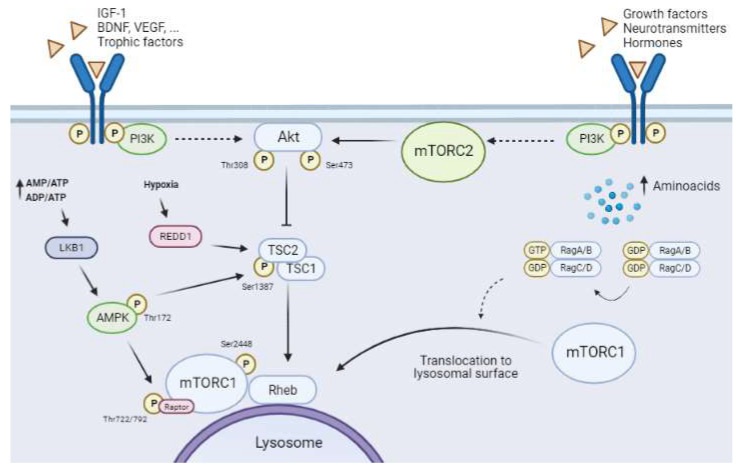
**Upstream mTORCs pathways.** Solid arrows show direct interactions between proteins and dashed arrows indicate the presence of other unrepresented mediators. The canonical PI3K/Akt pathway regulates mTORC1 activity through the binding of trophic and growth factors (BDNF, VEGF, IGF-1 among others) to RTK/GPCR-specific receptors. PI3K activation upregulates Akt by phosphorylation of Thr^308^. Full Akt activity requires the phosphorylation of Ser^473^ by mTORC2. Active Akt inhibits the tuberous sclerosis complex (TSC), which leads to induction of the GTPase Rheb, allowing activation of mTORC1. Amino acid availability leads to the translocation of mTORC1 to the lysosome membrane, which in turn allows its activation by Rheb. The AMPK pathway is activated in low-energetic states (AMP/ATP and ADP/ATP ratios increase). This activation requires the phosphorylation of Thr^172^ by LKB1. Active AMPK reduces mTORC1 activity through two distinct mechanisms, namely phosphorylation of Raptor at Ser^722/792^ and phosphorylation of TSC2 at Ser^1387^. Hypoxia induces an increase in REDD1, which downregulates mTORC1 by destabilizing TSC. Furthermore, mTORC2 activity is regulated by the presence of growth factors, neurotransmitters, and hormones, which all activate PI3K by binding to RTK/GPCRs.

**Figure 4 ijms-23-02814-f004:**
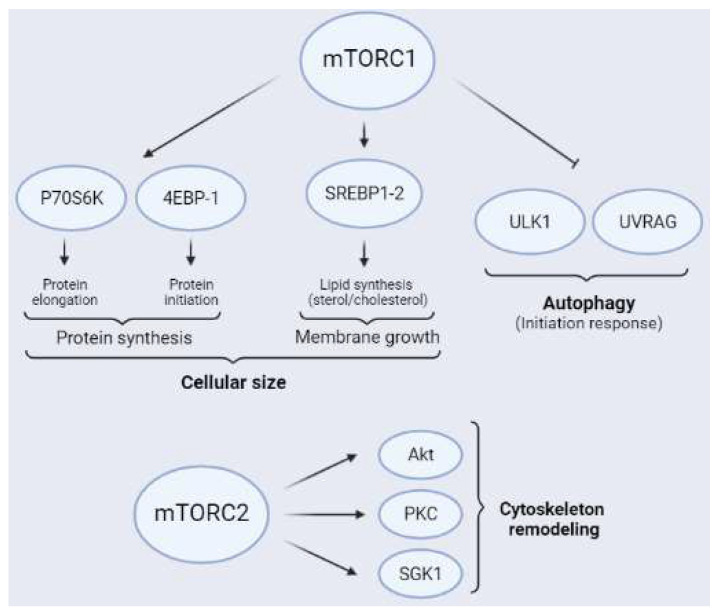
**Schematic representation of the main mTORC1 (above) and mTORC2 (below) substrates**. The main targets of mTORC1 related to protein synthesis are P70S6K and 4EBPs. Furthermore, mTORC1 regulates lipid synthesis through the transcriptional factor of lipogenesis SREBP1-2 and inhibits autophagy through ULK1 and UVRAG, the latter mainly in neurons. Additionally, mTORC2 is an activity modulator of Akt, PKC, and SGK1, all of which are related to cytoskeleton remodeling.

**Figure 5 ijms-23-02814-f005:**
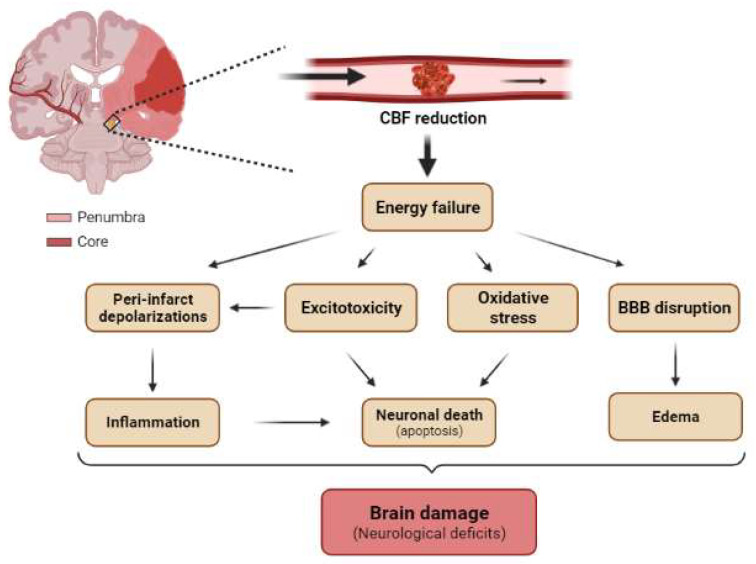
**Main pathological mechanisms of brain ischemia at tissue levels**. Schematic representation of a coronal section of the adult human brain with occlusion of MCA showing the core (red) and the penumbra (light red) regions. The reduction in cerebral blood flow (CBF) induces energetic failure at the tissue level, which promotes excitotoxicity, peri-infarct depolarization, oxidative stress, and BBB disruption. Secondarily, these pathological mechanisms triggers neuroinflammation, neuronal death (mainly by apoptosis), and cerebral edema, which in turn trigger tissue damage and neurological deficits.

**Figure 6 ijms-23-02814-f006:**
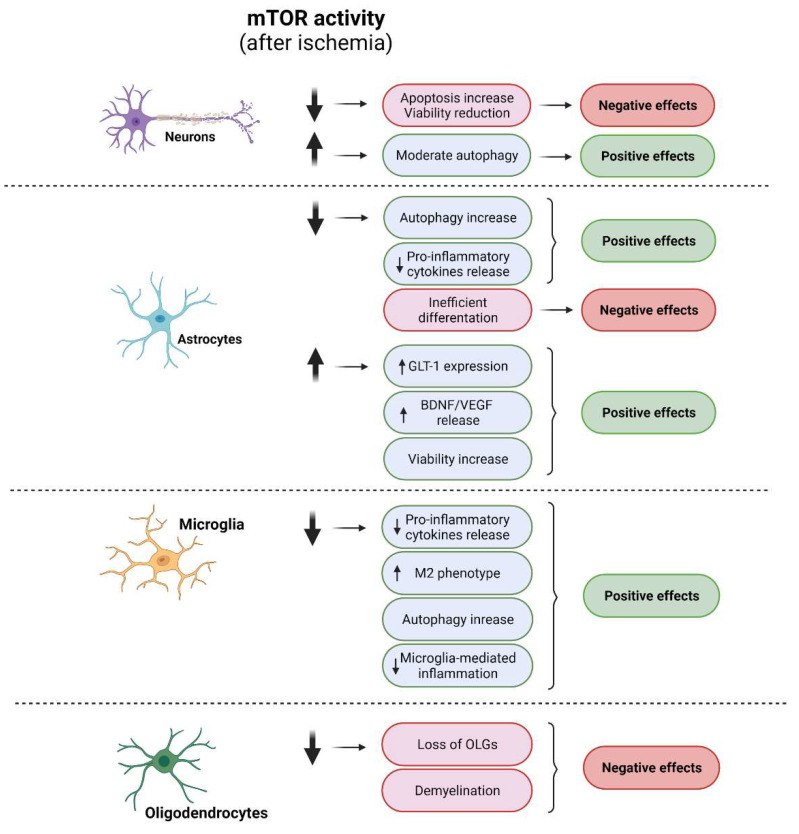
**Cellular effects of modulation of mTOR in ischemic conditions**. Diagram of the impacts of mTOR modulation in each neural cell type and their correlation with specific cellular processes. Positive effects are shown in green and negative effects in red. A diminution of mTOR activity after ischemia induces negative effects in neurons and OLG mainly related with a viability reduction, whereas an increment of this kinase activity triggers positive effects. In microglia, a reduction in mTOR activity shows positive effects via the induction of M2 phenotype and reduction in microglia-mediated inflammation. In astrocytes, either an activation of or reduction in mTOR activity shows positive effects.

**Figure 7 ijms-23-02814-f007:**
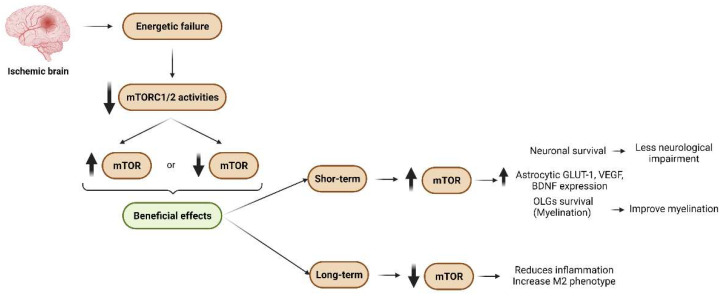
**Beneficial effects of upregulated or downregulated mTOR activity after ischemia**. Cerebral ischemia promotes energetic failure in the affected tissue, which induces reductions in mTORC1 and mTORC2 activities. Both an increment of and reduction in mTOR activity after ischemia result in beneficial effects. In the short term, activation of mTOR enhances OLGs and neuron survival, which improves myelination and reduces neurological impairment. Additionally, an increase in this activity in astrocytes induces GLUT-1 (that reduces excitotoxicity), VEGF, and BDNF expression. In the long term, the inhibition of mTOR activity reduces neuroinflammation and drives reactive microglia toward the M2 phenotype.

## Data Availability

Not applicable.

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
