# Peer review of "Dysregulation of mTOR Signaling after Brain Ischemia"

_ijms, 2022, doi:10.3390/ijms23052814_

Round 1

Reviewer 1 Report

The review “Dysregulation of mTOR signaling after brain ischemia” covers an interesting topic, which can contribute positively to the field. However, even if the work is well organized, in its present form the manuscript is difficult to understand and interpret, given the huge number of grammatical errors and inappropriate language. Hence, a thorough correction of English by a native English speaker is mandatory.

Several topics covered by the authors are superficially written, not supported by references, or supported by inappropriate references, especially in the first chapter “mTOR in Physiological Brain Conditions”.

As several sentences contain important and questionable statements a comprehensive study of the literature to which they refer should be carried out.

Just few examples:

- “Hypoxia induces an increment in Regulated in DNA Damage and Development 1 (REDD1) protein levels that activate AMPK activity” (REFs?). I understand that REDD1 activates AMPK, but it has been demonstrated that “the activation of AMPK controls REDD1 expression in response to prolonged hypoxia” (PMID:18953439).

- “It senses the availability of nutrients (glucose) and oxygen in the cell”. It is improper to define AMPK as an oxygen sensor, since it is still under debate whether AMPK activation is mediated by the hypoxic stimulus specifically or if it is a side effect of the energetic consequences due to hypoxia. In any case the right reference is missing.

- “It induces autophagy and facilitates the fusion of autophagosoma with lysosome, a key step of this process [62] (Check REF). In a nutrient-rich situation, active mTORC1 inhibits autophagy by phosphorylation of some proteins called: Unc-51-like kinase 1 (ULK1), UV radiation resistance-associated (UVRAG) or autophagy related gen 4 (Atg4) [63-66]”.  Here, I understand that mTOR phosphorylates just these three proteins, but they are not all the autophagy proteins phosphorylated by mTOR, the authors should edit the sentence. Also check the figure.

In conclusion this part must be completely revised, and accordingly to the text also the figures.

In the chapter “mTOR AFTER CEREBRAL ISCHEMIA”, the part of the text in which the authors address the effects of mTOR inhibition is very confusing, so the authors must rewrite this part, discuss, and make the necessary considerations on the evidence of literature.

Minor points.

- Many references are not included in the bibliography and appear as (Name et al. year) throughout the manuscript.

- The authors should standardize all the acronyms and the chapter titles.

Author Response

Dear Rewiewer

Thank you for your useful comments that have improve the manuscript. I answer your questions point by point (in red).

The review “Dysregulation of mTOR signaling after brain ischemia” covers an interesting topic, which can contribute positively to the field. However, even if the work is well organized, in its present form the manuscript is difficult to understand and interpret, given the huge number of grammatical errors and inappropriate language. Hence, a thorough correction of English by a native English speaker is mandatory.

A native English speaker has revised the text.

Several topics covered by the authors are superficially written, not supported by references, or supported by inappropriate references, especially in the first chapter “mTOR in Physiological Brain Conditions”.

We have added some new references through the text.

As several sentences contain important and questionable statements a comprehensive study of the literature to which they refer should be carried out.

Just few examples:

“Hypoxia induces an increment in Regulated in DNA Damage and Development 1 (REDD1) protein levels that activate AMPK activity” (REFs?). I understand that REDD1 activates AMPK, but it has been demonstrated that “the activation of AMPK controls REDD1 expression in response to prolonged hypoxia” (PMID:18953439).

We have explained these statements to clarify, since they were confusing: “A reduction of oxygen levels, as occurs under hypoxia, decreases cellular ATP levels by inhibiting oxidative phosphorylation and other metabolic programs. This scenario promotes an ATP/AMP imbalance, thus inducing AMPK activation and mTORC1 inhibition”. In addition, we have added a new section: 1.2.3. REDD1 and mTORC1.

“It senses the availability of nutrients (glucose) and oxygen in the cell”. It is improper to define AMPK as an oxygen sensor, since it is still under debate whether AMPK activation is mediated by the hypoxic stimulus specifically or if it is a side effect of the energetic consequences due to hypoxia. In any case the right reference is missing.

Exactly, this sentence was confusing since AMPK is not an oxygen sensor, hypoxia modifies cellular levels of AMP/ATP and ADP/ATP and that affects AMPK activity. We have clarified this subject  in the text and we have added  references.

- “It induces autophagy and facilitates the fusion of autophagosoma with lysosome, a key step of this process [62] (Check REF). In a nutrient-rich situation, active mTORC1 inhibits autophagy by phosphorylation of some proteins called: Unc-51-like kinase 1 (ULK1), UV radiation resistance-associated (UVRAG) or autophagy related gen 4 (Atg4) [63-66]”.  Here, I understand that mTOR phosphorylates just these three proteins, but they are not all the autophagy proteins phosphorylated by mTOR, the authors should edit the sentence. Also check the figure. In conclusion this part must be completely revised, and accordingly to the text also the figures.

We have edited the sentence and the figure.

In the chapter “mTOR AFTER CEREBRAL ISCHEMIA”, the part of the text in which the authors address the effects of mTOR inhibition is very confusing, so the authors must rewrite this part, discuss, and make the necessary considerations on the evidence of literature.

We have rewrite the section mTOR after cerebral ischemia to clarify it. In addition, we have modified the figure 5A to improve it  (now is Figure 5) and we have added the figure 6 (before Figure 5B) to show the cellular effects of modulation of mTOR in ischemic conditions.

Minor points.

- Many references are not included in the bibliography and appear as (Name et al. year) throughout the manuscript.

We have corrected this in the manuscript.

- The authors should standardize all the acronyms and the chapter titles.

We have standardized acronyms and chapter title.

Reviewer 2 Report

Dear authors!
The text of the manuscript needs to be edited.
The topic of the review that you propose is very relevant and fundamentally interesting. Nevertheless, in my opinion, it is necessary to make a deeper analysis of changes in mTORCs under conditions of ischemia, to put forward your own assumptions about the result of these changes. At the end of the manuscript, it is desirable to present in the figure the conclusions that you will draw regarding the role of mTORCs in ischemia-reperfusion conditions.

Author Response

Dear Reviewer

Thank you for your comments that have improve the manuscript. I answer your questions point by point (in red).

Dear authors!
The text of the manuscript needs to be edited.

A native English speaker has revised the text.

The topic of the review that you propose is very relevant and fundamentally interesting. Nevertheless, in my opinion, it is necessary to make a deeper analysis of changes in mTORCs under conditions of ischemia, to put forward your own assumptions about the result of these changes.

We have tried to clarify the conflicting data in the literature about mTOR and ischemia by rewriting and restructuring the text. In order to clarify, we have added Figure 6 showing schematically the effects of increase or decrease mTOR activity under ischemic conditions in each cellular type of CNS.

At the end of the manuscript, It is desirable to present in the figure the conclusions that you will draw regarding the role of mTORCs in ischemia-reperfusion conditions.

As you suggest, we have added a new figure (figure 7) that reflect the main conclusions of the manuscript.

Round 2

Reviewer 1 Report

Dear authors,

The manuscript "Dysregulation of mTOR signaling after brain ischemia" is greatly improved. However, before publication, given the complexity of mTOR activity in the different cell populations analyzed, it would be useful to add a detailed legend in Figures 6 and 7.

Authors should check for several typos throughout the manuscript, just two examples:

Lane 17:  as a therapeutic targets 

Lanes 647-648: understand

Author Response

Dear Rewiewer 1

Thank you for all of your suggestion that have been improve the manuscript. We have added a detailed legend in figure 6 and 7, we have checked the manuscript for typos, and we have corrected lane 17 and 647 as you suggest.

Reviewer 2 Report

Dear authors

I believe the manuscript has been greatly improved, although it could be made much more valuable by a deeper analysis of the consequences of mTOR activation or inhibition under ischemia/reperfusion conditions. For example, to clarify the role of survival of oligoderocytes depending on the type of fibrous
astrocytes. It is also more definite and clear to separate the short-term and long-term effects of mTOR inhibition. If you do this in the future, the work will acquire additional value.
At this stage, I believe the manuscript can be printed.

NOTES ON THE MANUSCRIPT

- The legend for Fig. 1 is in the wrong place.

- The legend for Fig. 4  “mTORC1 regulates lipid synthesis through the transcriptional factor of lipogenesis SREBP1-2 and inhibits autophagy”. However, as the authors write in the text, such a dependence is not always observed. Especially if we compare the effects in neurons and astrocytes. It is necessary to clarify the legend to Fig.4, indicating for which case this is true.

Author Response

Dear Rewiewer 2

Thank you for your suggestions. We have placed the figure legend in the right place. We clarify the legend to Figure 4.